

# Using regional relaxation experiments to understand the development of errors in the Asian Summer Monsoon

Gill M. Martin[1], José M. Rodriguez[1]

[1]Met Office, Exeter, EX1 3PB, UK

*Correspondence to*: Gill M. Martin (gill.martin@metoffice.gov.uk)

**Abstract.** We describe the use of regional relaxation ("nudging") experiments carried out in initialised hindcasts to shed light on the contribution from particular regions to the errors developing in the Asian Summer Monsoon. Results so far confirm previous hypotheses that errors in the Maritime Continent region contribute substantially to the East Asia Summer Monsoon (EASM) circulation errors through their effects on the Western North Pacific Subtropical High. Locally forced errors over the

Indian region also contribute to the EASM errors. Errors arising over the Maritime Continent region also affect the circulation and sea surface temperatures in the Equatorial Indian Ocean region, contributing to a persistent error pattern resembling a positive Indian Ocean Dipole phase. This is associated with circulation errors over India and the strengthening and extension of the westerly jet across southeast Asia and the South China Sea into the Western Pacific, thereby affecting the ASM circulation and rainfall patterns as a whole. However, errors developing rapidly in the deeper equatorial Indian Ocean,

apparently independently of the atmosphere errors, are also contributing to this bias pattern. Preliminary analysis of nudging increments over the Maritime Continent region suggests that these errors may at least partly be related to deficiencies in the convection and boundary layer parametrisations.

**Copyright statement**

## 1 Introduction

Monsoon systems, with their complex rainfall patterns and variability on a range of spatial and temporal scales, are emergent

phenomena whose simulation has proved a challenge for modelling systems over past decades. Model developers are making increasing use of a combination of various modelling techniques and sensitivity experiments of varying complexity in order



to try to understand the sources of common systematic biases in monsoon regions that have persisted throughout many generations of climate models (e.g. Bollasina and Nigam, 2009; Sperber et al., 2013; He et al., 2023).

Martin et al (2021) showed how various techniques and model configurations could be used to shed light on the progression
of bias development, local and remote drivers, and the role of atmosphere-ocean coupling. The tools and techniques allow close examination of the error development after initialisation, separation of the roles of local processes and remote teleconnections, identification of the contribution from errors developing in particular remote regions to those in the area of interest, and understanding of the role of atmosphere–ocean coupling. One of these techniques is the use of regional relaxation ("nudging") experiments, where certain model variables are relaxed back towards reanalyses in particular regions and the
effects on the errors developing in another region are examined. Martin et al. (2021) described the use of such nudging experiments to analyse the contribution to errors arising in the East Asia Summer Monsoon (EASM) and South Asian Summer Monsoon (SASM) in the Met Office models from those arising over the Maritime Continent, Philippines and South China Sea and the Equatorial Indian Ocean. These experiments, which included both free-running climate simulations and initialised numerical weather prediction (NWP) hindcasts, were carried out using atmosphere-only configurations. While these give a
good indication that it is mainly the atmosphere in the Philippines, Maritime Continent and Indian Ocean regions that drives the errors in the EASM and SASM, there were also indications that some errors (e.g. in the Equatorial Indian Ocean) have at least some of their origin in the ocean. Coupled feedbacks exacerbate such errors and also make it difficult to unambiguously identify misrepresentation of either atmosphere or ocean processes.

In order to start to unpick the contribution from coupled processes, and also to increase the sample size, regional relaxation
ensemble experiments have been carried out in both coupled NWP and seasonal hindcasts using the Met Office Global Coupled model. Such experiments have been carried out in other modelling systems and for other purposes. These include identification of large-scale, remote or global influences on unusual or extreme events in particular locations (e.g. Jung et al., 2010; Maidens et al., 2019; Knight et al., 2017, 2021), reducing internal variability so that shorter simulations can be used to quantify the impact of changes to model parameterizations (e.g. Lohmann and Hoose, 2009; Kooperman et al., 2012), for understanding
the role of model errors in representing modes of variability on the skill associated with their teleconnections (e.g. Douville et al., 2011; Beverley et al., 2021; Martin et al., 2023) and examining the influence of atmospheric circulation or temperature anomalies in a particular region on forecast errors in another (e.g. Rodriguez et al., 2017; Rodriguez and Milton, 2019; Dias et al., 2021; Martin et al., 2021; Mayer et al., 2023).

Few studies have used nudging to examine the role model errors in one region on the development or "spin-up" of model
errors in another location. This is best achieved by applying regional relaxation to reanalyses in an initialised system where the effect of preventing such errors from developing from the start can be investigated. Since unpicking model spin-up characteristics involves analysing the errors on short timescales, it is helpful to use a hindcast ensemble carried out over many years and with multiple ensemble members per start date, in order to improve sampling and reduce noise. Following on from the study of Martin et al. (2021), which used both seasonal and NWP hindcasts in a seamless modelling system to analyse the
spin-up of errors in the Met Office models, in the present study we apply regional relaxation towards reanalyses in both systems





over specific regions that were identified as likely locations where model errors were influencing the wider simulation of the Asian summer monsoon (ASM). We then examine how correcting the atmospheric temperature and circulation over these regions affects the development of atmosphere and ocean errors across the ASM region as a whole, with specific focus on the equatorial Indian Ocean where the Met Office and other climate models show a persistent systematic bias in sea surface

temperatures.

The paper is arranged as follows: in Section 2 we describe the data and methods used, while Sect. 3 documents the results of the various experiments. We summarise our discussion in Sect 4.

## 2 Data description and methods

Hindcast ensembles have been generated using 5 relaxation regions, plus one where global relaxation to reanalyses is applied.

In GloSea5 nudging experiments, horizontal winds and potential temperatures are nudged back to ERA-Interim reanalyses (ERA-I; Dee et al., 2011) with a 6-hourly relaxation timescale at all model levels up to the top of the atmosphere. Note that the reanalyses extend only to 65km (approximately model level 80) so above this level a vertical extrapolation is applied. A 10° buffer zone around the relaxation subdomain is applied in which the nudging increments are exponentially damped to zero. This helps to ensure a smooth transition between the nudged and free-running parts of the simulation. The GloSea5

configuration is as at the 2018 operational hindcast (matching the dataset analysed in Martin et al., 2021) and uses Global Coupled model version 2 (GC2; Williams et al, 2015). Hindcasts are carried out for 4 start dates (25 March, April, May and June) over 23 years (1993-2015) and with 10 ensemble members each. We compare our results against ERA-I for winds, the Global Precipitation Climatology Project pentad dataset version 2.2 (GPCP v2.2; Adler et al., 2003) for precipitation, and National Oceanic and Atmospheric Administration (NOAA) Optimum Interpolation Sea Surface Temperature v2 (OISSTv2;

Reynolds et al., 2007). We average the ensemble mean precipitation, winds and sea surface temperatures (SSTs) into pentads, and average both the model and observational fields over the hindcast period (1993-2015), in order to reduce the effects of internal variability.

Coupled NWP hindcasts use Global Coupled model v3.0 (GC3; Williams et al., 2017) and are carried out at N320ORCA025 resolution (~40km atmosphere and ~25km ocean in the tropics; denoted CNWP-N320), with one ensemble member initialised

each day and run for 30 days of the year 2020. Forecast validity times between 30 June to 31st August (62 cases) are used to construct composites. SSTs and ocean mixed layer depths are compared against Forecast Ocean Assimilation Model (FOAM) Ocean Analysis (Blockley et al., 2014; Waters et al., 2015) while atmospheric fields are verified against Met Office Unified Model (MetUM) operational analyses.

The nudging regions are shown in Fig. 1. The Philippines, Indonesia and Maritime Continent regions are the same as those

used by Martin et al. (2021; their Fig. 14). These regions were chosen based on analysis which indicated that the Maritime Continent region may be influencing the development of errors in the EASM, and that the Philippines and Indonesia regions may contribute both independently and jointly. For the SASM region, previous published studies (Levine and Martin, 2018; Levine et al., 2013; Marathayil et al., 2013) indicate that the Equatorial Indian Ocean plays a role but also that many of the



errors are locally driven. Two additional regions are therefore included, covering India [10°-25°N 60°-95°E] and the Equatorial

Indian Ocean [10°S-5°N 50°-95°E].

Assuming a linear response, the difference between the control and the "nudged" simulations (Control minus Nudged) gives an indication of the role played by the nudged region in the errors that occur in the control in other locations, while the difference between the regionally-nudged and globally-nudged simulations (Nudged minus Nudge All) gives an indication of the role played by areas outside the chosen region. We also compute the difference between the nudged simulations and the

observations to show how the error patterns develop when the winds and temperatures in the region of interest are constrained to the reanalyses.

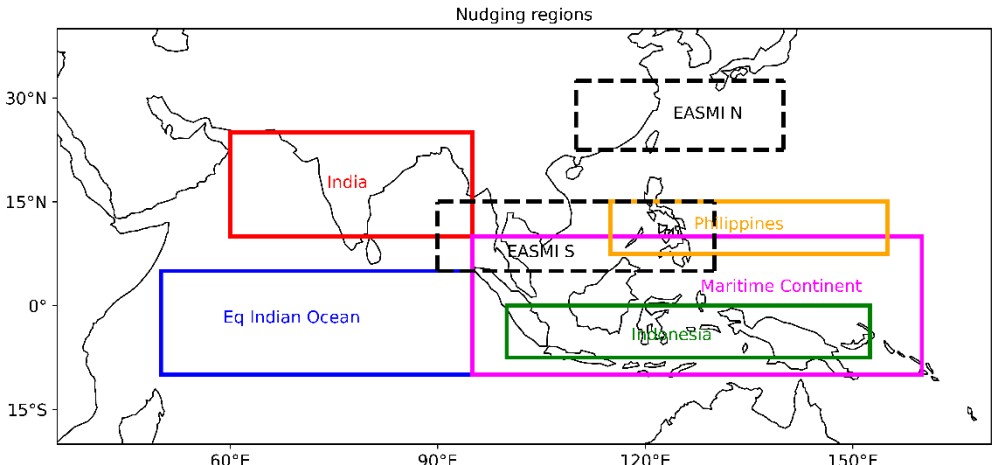

**Figure 1: Regions where 6hr relaxation towards ERA-I reanalyses is applied to atmospheric temperature and horizontal winds. India: [10°-25°N 60°-95°E]; Eq. India Ocean: [10°S-5°N 50°-95°E]; Philippines: [7.5°-15°N 115°-155°E]; Indonesia: [7.5°S-0°N**

**100°-152.5°E]; Maritime Continent: [10°S-10°N 95°-160°E]. Black dashed boxes indicate the regions used to calculate the East Asia Summer Monsoon Index (EASMI): EASMI N [22.5–32.5°N, 110–140°E] and EASMI S [5–15°N, 90–130°E].**

## 3 Evolution of model errors in the ASM region

### 3.1 ASM errors in seasonal hindcasts

Figure 2 (left) shows the climatological development of rainfall and wind errors in the seasonal hindcast ensemble, averaged

over the first, third and fifth pentads following initialisation on 25 June. Excess precipitation occurs in the Equatorial Indian Ocean (EIO), eastern Bay of Bengal (BoB), South China Sea (SCS) and western Pacific in the first pentad after initialisation, with deficient precipitation over India and around the Maritime Continent. Diverging wind anomalies are seen along the west coast of India which develop into an anticyclonic anomaly that is associated with a weakening of the SASM trough. Negative rainfall anomalies develop over northwest India that are likely to be associated with an underestimation of the number of

monsoon lows and depressions and the extent of their westward progression across northern India (Levine and Martin, 2018). The westerly flow accelerates across the BoB and southeast Asia, converging with southerly anomalies diverging from the Maritime Continent region. This promotes further rainfall in the SCS and western Pacific, creating a positive feedback that




further promotes extension of the westerly jet across the SCS and the Philippines into the western Pacific. As the hindcasts progress, the Western North Pacific Subtropical High (WNPSH) weakens and shifts towards the east, reflected as an anomalous

cyclonic pattern over the western Pacific and northeasterly anomalies over southeast China. Positive rainfall errors over the Indian Ocean to the south of the Indian peninsula are associated with the anomalous north/northwesterly winds from the Arabian Sea converging with the anomalous easterly winds from the eastern equatorial Indian Ocean (EEIO). After around 15 days, the patterns of rainfall and wind errors closely resemble the climatological errors in the free-running simulations (see Martin et al., 2021; their Fig. 1).


**Figure 2: Errors in (left) 850 hPa winds (vs ERAI) and rainfall (vs GPCP) and (right) SST (vs OISSTv2) averaged over the first, third and fifth pentad after initialisation in GloSea5 operational hindcast initialised on 25 June 1993-2015.**



At the start of the hindcast, warm SST anomalies are apparent in the SCS and western Pacific (Fig. 2, right). These illustrate the differences between the OISSTv2 observations and the GloSea5 initialisation from ocean and sea-ice analysis using

GloSea5 Global Ocean 3.0 driven by ERA-I and using the NEMOVAR data assimilation scheme (discussed later in this section). Over the first 15-20 days of the hindcast, cold anomalies develop around the Indonesian islands and spread westwards across the EIO in association with increasing southeasterly anomalies along the Sumatran coast and diverging 850 hPa wind anomalies and a negative rainfall error across the whole of the Maritime Continent. Cold SST errors are also seen along the northern and western Arabian Sea coasts, while warm SST errors develop in the western equatorial Indian Ocean and central

and eastern Arabian Sea. Combined with the cold SST errors in the eastern EIO, these form the east–west dipole error pattern seen in the climatological SST errors (see Martin et al., 2021; their Fig. 2).

Martin et al. (2021) suggested that errors over the Philippines and the Maritime Continent regions play a joint role in the EASM circulation errors via weakening and displacement of the WNPSH. Our "Nudge Philippines" and "Nudge Indonesia" experiments (not shown) confirm the results of Martin et al. (2021) that "the 'Indonesia' region promotes westerly anomalies

extending from the South Asian monsoon westerly jet across the Philippines into the western Pacific, while the 'Philippines' region promotes additional acceleration of these westerly winds as part of an anomalous cyclonic circulation that includes northeasterly anomalies over southern China. Both regions promote excess rainfall over the eastern SCS and the western Pacific". Figure 3(left) shows the contribution from errors in the "Maritime Continent" (MC) region to the ASM circulation and rainfall errors, and the effects on the total errors of nudging this region back to reanalyses. Again similar to Martin et al.

(2021), we see that errors in this region drive acceleration of the westerly flow across the SCS and Philippines into the West Pacific and the development of cyclonic anomalies over southeast China. When the atmospheric circulation and temperature errors in the MC region are removed (Fig. 3(right)) the circulation and rainfall errors over southeast Asia, the SCS, Philippines and West Pacific and over China are all greatly reduced. However, easterly anomalies that form part of the northern flank of the anomalous cyclonic circulation over the West Pacific in the Control (Fig. 2(left)), extending from the Central Pacific and

accelerating south of Japan across the East China Sea and into southern China, are still present when the MC region is nudged, suggesting that these errors originate elsewhere.

Martin et al. (2021) further suggested that the Indian Ocean errors were also related to those over and around the Maritime Continent islands, whereby south-easterly wind errors along the Sumatran coast develop westwards across the EEIO and drive increased ocean upwelling and colder SSTs in the east, resulting in warmer SSTs in the west. However, comparison of Fig.

2(left) and Fig. 3(right) shows that, while there is a reduction in the errors over the eastern EIO when the MC region is nudged, the errors over the SASM region are largely unchanged while those over the central and northern EIO are worsened. This suggests that, while the errors over and around the MC islands feed into the EIO region, they are largely developed and exacerbated locally via coupled feedbacks which promote an east-west dipole SST anomaly with positive feedback on the equatorial winds.

Figure 4(left) shows the contribution of the EIO region to the error development. Comparison with Fig. 2 suggests that much of the ASM circulation error pattern is at least partly associated with errors developing over the EIO, particularly the





accelerated westerly flow across the BoB, SCS and Philippines into the West Pacific and the cyclonic anomalies (weakening and displacement of the WNPSH) affecting the EASM. Figure 4(right) shows how these errors are reduced when nudging of winds and temperatures is applied in the EIO region. The divergence of the 850 hPa winds along the Indian coast remains, however, along with the developing deficit in rainfall there, suggesting that these may be more locally driven. In addition, the easterly anomalies in the Western Pacific feeding into southeast China are also still present when the EIO region is nudged, suggesting again that these originate elsewhere.

**Figure 3: (left) Climatological differences in 850 hPa winds and rainfall, averaged over the first, third and fifth pentads after initialisation on 25 June 1993-2015, between "Nudge MC" experiment (nudging applied over "Maritime Continent" (MC) region [10°S-10°N 95°-160°E]) and the GloSea5 operational hindcast (Control), indicating the influence on the control simulation of**



**errors developing over the MC region; (right) errors in 850 hPa winds (vs ERAI) and rainfall (vs GPCP) in the hindcast with nudging applied over the MC region. The MC region is shown by the red box.**


**Figure 4: As Fig. 3 but for the hindcast with nudging applied over the Eq. India Ocean (EIO) region [10°S-5°N 50°-95°E], shown by the red box.**



Interestingly, the excess rainfall over the EIO itself (particularly north of the Equator) remains present when the winds and temperatures in that region are nudged back to reanalyses. This error appears to originate just to the north of the EIO nudging region and spread southwards. Figure 5(left) shows the results for the nudged "India" region. This indicates that circulation and temperature errors over the Indian region contribute to the accelerated westerlies across southeast Asia and the SCS and to the cyclonic anomalies affecting the EASM. The plot also suggests that errors in the Indian region drive anomalous outflow

into the central and eastern EIO and reduced rainfall around the southern tip of India, so that when nudging is applied to the Indian region the excess rainfall in the region just south of the box is increased (Fig. 5(right)). This implies that errors developing in both the Indian region and the EIO region provide a compensating drying in this region against the increasing rainfall there that is perhaps driven locally. However, since the area in which this important bias seems to develop lies mainly between the two nudging regions and could therefore be affected by boundary effects (despite the 10° tapering), this

preliminary conclusion needs to be confirmed using further experiments. The role of the western Arabian Sea is also not clear; it is possible that the Indian nudging region is too small. In future experiments we may extend this region westwards to 50°E and southwards to 5°N.





**Figure 5: As Fig. 3 but for the hindcast with nudging applied over the "India" region [10°-25°N 60°-95°E], shown by the red box.**


Figure 6 shows that the cold SST errors that develop around the MC islands are largely eliminated when the atmospheric circulation and temperatures in that region are nudged back to reanalyses, indicating that the atmosphere is driving these local SST errors. However, while the cold SST error in the EEIO just south of the Equator is also eliminated when the MC region is nudged, most of the SST errors over the Indian Ocean as a whole remain despite the reduction in wind errors over the central



and eastern Indian Ocean. Further, we see in Fig. 7 that nudging the EIO region also reduces the cold SSTs around the MC islands and the EEIO.



**Figure 6:** (left) Climatological differences in SST, averaged over the first, third and fifth pentads after initialisation on 25 June 1993-2015, between "Nudge MC" experiment (nudging applied over Maritime Continent (MC) region [10°S-10°N 95°-160°E]) and the GloSea5 operational hindcast (Control), indicating the influence on the control simulation of errors developing over the MC region; (right) errors in SST (vs OISSTv2) in the hindcast with nudging applied over the MC region, shown by the red box.



**Figure 7: As Fig. 6 but for the hindcast with nudging applied over the Eq. India Ocean (EIO) region [10°S-5°N 50°-95°E] shown by the red box.**

Figure 7(left) also suggests that the circulation anomalies over SE Asia and the western Pacific, that are associated with errors over the EIO, are also associated with local cooling of SSTs in the northern Arabian Sea, western BoB, SCS and West Pacific. Figure 6(left) also demonstrates the role of atmospheric errors in the MC region in driving cooler SSTs in the SCS. Circulation and temperature errors in the India and Indonesia regions also promote cooler SSTs in the SCS and West Pacific via the accelerated westerly flow across those oceans (not shown). Hence when any of these regions is nudged back to reanalyses, warmer SST errors than in the Control develop in the SCS and West Pacific during June, and there is far less widespread cooling than is seen in the Control (Fig. 2).



When the winds and temperatures in the EIO region are nudged to reanalyses, the SST errors within this box to the south of the Equator are much reduced (Fig. 7(right)). This indicates that these ocean surface temperature errors are driven, at least in part, by the atmosphere. However, the SSTs in this box to the north of the Equator, and those in the Western EIO (WEIO), develop a cold bias when the atmosphere is nudged. These are associated with both the tendency for excess rainfall to develop to the south of India (see Fig 2) and the other circulation and rainfall errors that develop around this region when it is nudged (see Fig. 4), both of which require further investigation.

In Figure 8 we show the effects on the East Asia Summer Monsoon Index (EASMI; see Wang et al., 2008) of nudging the various remote regions in hindcasts initiated on 25th April, May and June. The EASMI measures the circulation strength via the difference in the zonal wind at 850 hPa (U850) between [22.5–32.5°N, 110–140°E] and [5–15°N, 90–130°E] (see EASMI N and EASMI S boxes on Fig. 1). In the Control, the EASMI decreases rapidly after initialisation in all hindcasts, indicating the weakening and displacement of the WNPSH. The two components reveal that this is driven mainly by the increasingly excessive westerly flow in the southernmost box and a rapidly developing easterly error in the northernmost box. The MC nudging region overlaps with part of the EASMI S region, so it is perhaps not surprising that this component is improved when this region is nudged. However, we see from Fig. 8 that the EASMI N component is also improved, particularly when nudging the MC region. Indeed, for all 4 start dates the EASMI in the Nudge MC hindcasts agree well with ERA-I. Noticeable improvement is also seen in the Nudge EIO and Nudge India hindcasts compared with the Control.

These results suggest that preventing wind and temperature errors from developing over these various nudging regions reduces a characteristic systematic error of the EASM in Met Office models that is associated with a lack of northward advancement of the Meiyu rain band and a deficit in the seasonal mean rainfall (e.g. Zhang et al., 2020; Martin et al., 2020). Analysis in Sect. 3.2 reveals that deficient ascent over the MC region from the outset is associated with anomalous ascent over the SCS and West Pacific, reducing the boreal summer Hadley circulation. Preliminary analysis of temperature increments from the regional nudging experiments (Fig. 9) indicate that the model tends to produce insufficient warming (or too much cooling) around the freezing level and not enough sub-cloud layer cooling or enough warming above, so that the nudging increments oppose these errors. This may suggest that the errors arising in the MC region are at least partly related to deficiencies in the convection and boundary layer parametrisations; this will be investigated in more detail in future work.

The analysis above sheds light on the influence of particular regional atmospheric errors on the circulation, rainfall and SSTs elsewhere in the region. However, it is also clear that there are interactions and feedbacks between the errors in the different regions which combine non-linearly to produce the overall biases in the Control run. It is also important to recognise that nudging only winds and atmospheric temperatures does not fully constrain the distribution of clouds and rainfall, which themselves influence the surface heat budget and hence the SSTs. In addition, errors driven by the ocean model will also contribute to those seen in the coupled system.



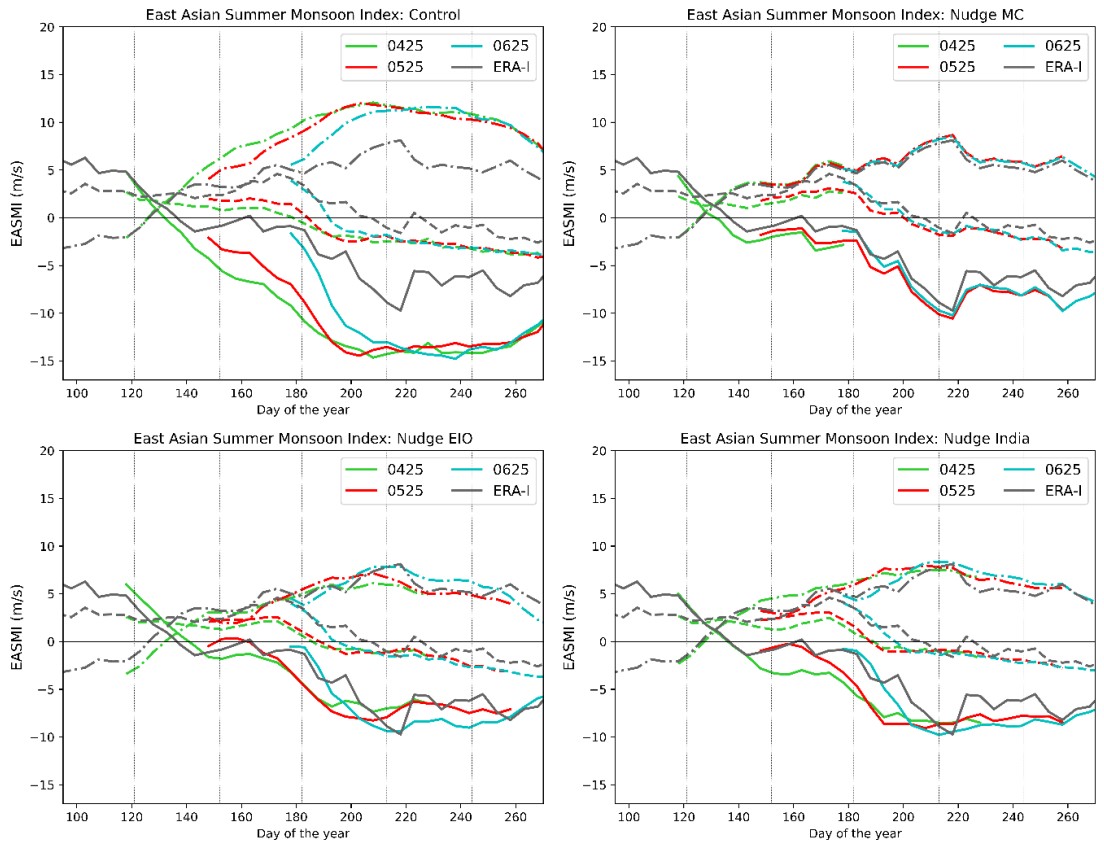

**Figure 8: East Asia Summer Monsoon circulation index (EASMI: 850 hPa zonal wind difference (22.5–32.5° N, 110-140° E) – (5–15° N, 90–130° E); see Wang et al., 2008) in Control and Nudged runs with nudging to winds and temperature applied over MC, EIO and India regions, initialised on 25 April, May and June 1993-2015 (0425, 0525, 0625 respectively), compared with ERA-I. Dashed = northern box; dot-dashed = southern box, solid = EASMI. Vertical lines indicate the calendar months (1 June is day 151 in a no-leap Gregorian calendar).**

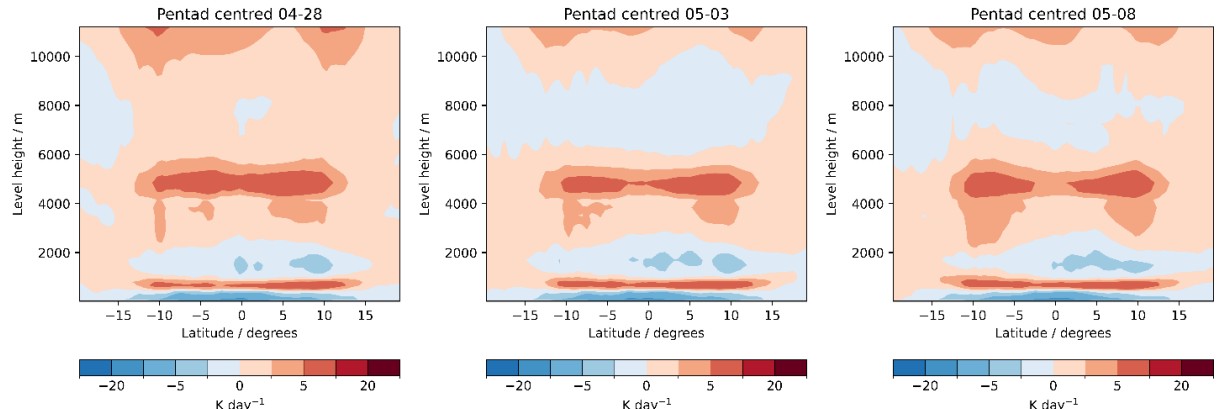

**Figure 9: Nudging increments to temperature (K day⁻¹) for the first 3 pentads after initialisation on 25 April, averaged over 10-member ensembles and from 1993-2015. Latitude-height cross sections across MC nudging region (95°-160°E, 10°S-10°N, with 10° damping zone on all sides).**





Examination of the SST errors (against OISSTv2 observations) in the globally nudged simulation (Fig. 10, right) reveals that
260  the characteristic dipole error pattern in SSTs across the EIO develops even when the atmospheric temperatures and winds are
relaxed globally to reanalyses, albeit to a lesser extent than in the Control (Fig. 2). Warm SST errors are seen in the western
and southwest Indian Ocean and particularly in the northwest Arabian Sea, and also around the coasts of eastern China, South
Korea and Japan, while a widespread cold SST bias develops across the western/central Pacific and to the north of Australia.
Comparison with the 850 hPa wind and precipitation errors from the same hindcasts (Fig. 10, left) reveals a close relationship
265  between some of the positive rainfall errors and the developing cold SST errors, e.g. in the EIO to the south of the Indian
peninsula and in the West Pacific to the east of the Philippines. These rainfall errors are characteristic of typical model biases
in the MetUM that have been previously documented in atmosphere-only simulations (e.g. Keane et al., 2019; Johnson et al,
2016; Martin et al., 2010). Their persistence and effect on the local circulation, despite the nudging back to reanalyses, suggests
that their origin is in the atmospheric model physical parametrisations, particularly the convection scheme (Bush et al., 2014).
270  It is possible that the more widespread changes in SST may also be related to the differences between the GloSea5 Ocean and
Sea Ice Analyses (see details in MacLachlan et al., 2015) that are applied as initial conditions to the ocean model in the
hindcasts and the satellite-based SST retrievals in OISSTv2. A comparison between the SSTs on the initialisation date (25
June) with OISSTv2 SSTs on the same day, averaged over the period 1993-2015, reveals systematically warmer SSTs in the
reanalyses than in OISSTv2 over much of the western Pacific including the East China Sea and the Sea of Japan, and
275  systematically cooler SSTs over much of the northern and equatorial Indian Ocean with warmer SSTs further south (Fig. 11).
Very similar difference patterns are seen on other start dates through the monsoon season (not shown). In future work we will
investigate the role of the ocean initialisation on the development of ASM errors.

## 3.2 Emergence of Indian Ocean SST errors in coupled NWP hindcasts

We now focus on the emergence of Indian Ocean SST errors mentioned above, since these are a key and persistent systematic
280  error in Met Office and other modelling systems (e.g. Mayer et al, 2023). Following evidence in Martin et al. (2021) that
similar errors develop in our coupled numerical weather prediction initialised hindcasts, and that they are largely insensitive
to horizontal resolution, we analyse coupled NWP hindcasts using Global Coupled model v3.0 (GC3; Williams et al., 2017)
carried out at N320ORCA025 resolution (denoted CNWP-N320), with one ensemble member initialised each day and run for
30 days of the year 2020. Forecast validity times between 30 June to 31st August (62 cases) are used to construct composites.
We confirm that the development of SST biases in the CNWP-N320 (GC3) initialised hindcasts (not shown) resembles that in
the GloSea5-GC2 seasonal hindcasts (Fig. 2): by forecast lead time 30, the characteristic dipole error pattern, resembling that
of the Indian Ocean Dipole (IOD; Saji et al., 1999) positive phase is established, with cold biases around the Indonesian islands
and a strong warm bias in the WEIO.  To consider in more detail the evolution of the SST biases in the Indian Ocean, we
centre our analysis in two regions representing the EEIO and WEIO, where we calculate area-average values of relevant ocean
and atmospheric diagnostics over the boxes shown in Fig. 13.





**Figure 10: Errors in (left) 850 hPa winds (vectors) and rainfall (shading) and (right) SSTs from the first, fifth and ninth pentad after initialisation in globally nudged seasonal hindcasts initialised on 25 June 1993-2015. Wind errors are relative to ERA-I, rainfall errors against GPCPv2 and SST errors against OISSTv2.**



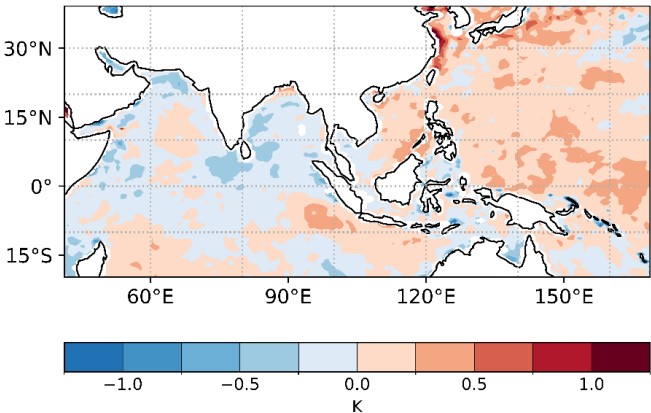


**Figure 11: SST difference between the GloSea5 initialisation (day 1 of operational hindcast SSTs averaged over 25 June 1993-2015) from ocean and sea-ice analysis using GloSea5 Global Ocean 3.0 driven by ERA-I and using the NEMOVAR data assimilation scheme and OISSTv2 observations on 25 June averaged over 1993-2015.**

We first present the evolution in forecast lead time of biases in the eastern tropical Indian Ocean region (see Figure 12, where errors for CNWP-N320 hindcasts are shown as solid lines). Figure 12a shows SST and near-surface (10m) wind speed errors, averaged over all forecasts with validity times between 30 June and 31 August. Compared to FOAM analysis, the hindcasts show an SST changing error that starts as a spin-up positive bias, lasting 14 days and later turning into a long-term negative bias, with a small negative trend.  There is evidence that the atmosphere is forcing the long-term ocean error in this region.

Three different mechanisms, with a strong diurnal component, set up the formation of temperature gradients in the upper ocean and determine the value of SSTs:  the absorption of insolation, the heat loss to the atmosphere by radiation and conduction, and the amount of sub-surface turbulent mixing, driven by surface wind, which erodes any thermal stratification, brings colder subsurface water to the surface and reduces SST values.  Surface wind plays a significant role in driving the SST bias in the region, especially at the longer term (after 15 days). This can be seen in Fig. 12a where the surface wind and SST errors are

highly anti-correlated. This is consistent with the findings of Gupta et al. (2022) for the NCMRWF coupled model based on GC2. Accordingly, Fig. 12b confirms a deepening of the mixed layer in the model after a spin-up period of 15 days, consistent with the strong-wind bias during that period and again consistent with the findings of Gupta et al. (2022). The error development in the mixed layer depth is highly correlated with the surface wind speed error, with a lag of approximately 1 day. At long lead times, most of the wind speed errors correspond to the strong westerly flow bias in the area, mentioned in

section 3.1 (see agreement between blue and cyan lines). Net surface heat is the atmospheric driver of the SST warm bias during the spin-up phase (the first 14 days, as shown Fig.12c). The main contributions to that excessive flux are positive biases in the short-wave radiation and latent heat. As the strong-wind error develops at longer lead times, the latent heat bias becomes negative and further contributes to the decrease of SSTs.  Overestimated precipitation in the area also contributes to the long-term cold SST bias, by affecting the local surface SW flux.  Precipitation and SW flux errors are highly anticorrelated in the




area.  As the excessive precipitation increases after the spin-up phase, the SW flux decreases, reducing the absorption of solar

radiation by the ocean and increasing the cold SST bias.

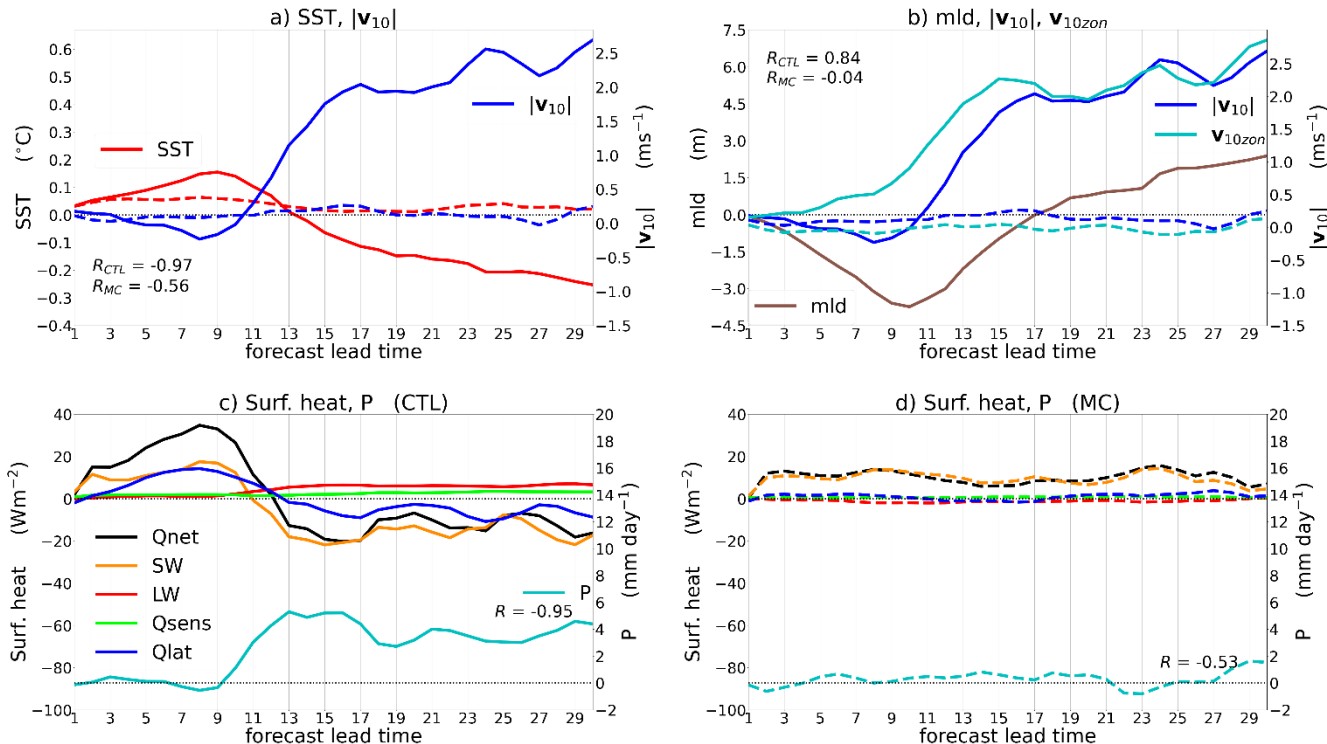

**Figure 12: Mean biases over the eastern tropical Indian Ocean region (90° - 97.5° E, 7.5° - 15° N) in CNWP-N320 hindcasts (CTL,**
**solid lines) and hindcasts with nudging applied to the Maritime Continent region (MC, dashed lines), plotted against forecast lead**
**time (days).  (a)  SST error (red, °C) and 10-m wind speed error ($|\mathbf{v}_{10}|$, blue, ms$^{-1}$). Pearson correlation coefficients (R) between SST**
**and surface wind are also shown. (b) Mixed-layer depth error (brown, m), 10-m wind speed error ($|\mathbf{v}_{10}|$, blue, ms$^{-1}$) and its zonal**
**component ($\mathbf{v}_{10zon}$, cyan). Correlation coefficients between mixed-later depth and surface wind speed are also indicated.  (c)**
**Precipitation (P, cyan, mmday$^{-1}$) and surface net heat flux errors in CTL hindcasts (Qnet, black, Wm$^{-2}$) and its various**
**components: SW radiation (amber), LW radiation (red), sensible heat (green) and latent heat (blue).  The correlation coefficient R**
**between precipitation and surface SW radiation is shown on the bottom right. (d) Same as (c), but for hindcasts with MC nudging.**
**SST and mixed-layer depth errors are calculated against FOAM analysis, and 10-m wind, surface heat and precipitation errors,**
**against MetUM analysis.**

We have carried out a series of CNWP-N320 sensitivity experiments using the nudging/relaxation methodology described in

section 2. We present here the remote influence of the Maritime Continent (MC) on the development of biases in the EIO area.



The MC region is defined as in section 3.1. Nudging atmospheric temperature and wind over the Maritime Continent greatly reduces the development of wind and SST biases in the region (see dashed lines in Figure 12). The regional surface wind error is very small and does not seem to be driving the SST bias (wind speed-SST error correlations are much smaller in this case). The area-average SST shows a small warm bias that is caused by an excessive surface SW radiation in the region (see Figure 340    12a and 12d).

Errors in surface wind play an essential role in the development of the model's SST bias in the eastern tropical Indian Ocean region. We now explore how they can be linked to the large-scale circulation errors in the model. Figure 13a-c shows 30 June – 31 August mean circulation errors from CNWP-N320 hindcasts at lead times of 5, 15 and 30 days. It displays 500 hPa vertical wind (shading) and 250 hPa divergent horizontal wind errors (streamlines) in the tropical Indian Ocean and West 345    Pacific. This shows suppression of vertical motion in places of upper-level convergence and excessive vertical motion in places of upper-level divergence. By day 30, the vertical motion errors show a pattern in agreement with the precipitation errors shown in section 2, with a deficient upward motion in the Maritime Continent and adjacent ocean, the Indian Peninsula and the western tropical Indian Ocean and zones of excessive lift circumscribing the Maritime Continent and in the EIO. They form two large erroneous cells in the region, going from the place of excessive upward motion in the West Pacific to the 350    Maritime Continent (in a sort of erroneous Hadley cell), and from the EIO to the Indian Peninsula and the Eastern Arabian Sea. These erroneous circulation cells connect surface wind biases in the region, such as the excessive south westerlies in the EEIO with the EIO region of excessive precipitation, and the strong easterly flow in the eastern tropical Indian Ocean with the Maritime Continent. When the CNWP-N320 hindcasts are globally nudged, the circulation errors are greatly reduced, but a weaker erroneous cell, connecting the zone of excessive lift in the EIO and the Indian Peninsula, remains (See Fig. 13d-f).

We now consider the evolution of biases in the WEIO region (box 2 in Fig. 13). The SST shows a warm bias that grows monotonically with lead time (Fig. 14a). SST and surface wind speed biases are now slightly correlated (instead of anticorrelated) and there is no correlation between surface wind and mixed layer errors. Note that, for this region, the wind speed error is coming from the meridional wind component (the zonal wind error is negligible). On the other hand, although there is a slight excessive net surface heat flux after 9 days, it is too small to produce such a large positive SST bias (see Fig. 360    14c). Nudging winds and atmospheric temperature over the MC region do not change qualitatively this picture: the SST error evolution is similar, despite slight differences in the surface winds and heat fluxes. This situation remains the same, even in the case of globally nudged hindcasts (not shown). These results suggest that local surface wind speed and surface heat fluxes are not driving the SST bias in the WEIO region, and that there is an oceanic transport component to the error.






**Figure 13: Mean 500 hPa vertical wind errors (shadings, Pa s $^{-1}$) and 250-hPa horizontal divergent wind errors (streamlines) for NWP-N320 hindcast (CTL, left) and hindcasts with global nudging (GBL, right). Composite errors are shown at lead times of 5, 15 and 30 days. Errors are calculated against UM analysis. Boxes indicate the regions over which various quantities are averaged in Fig.s 12 and 14.**



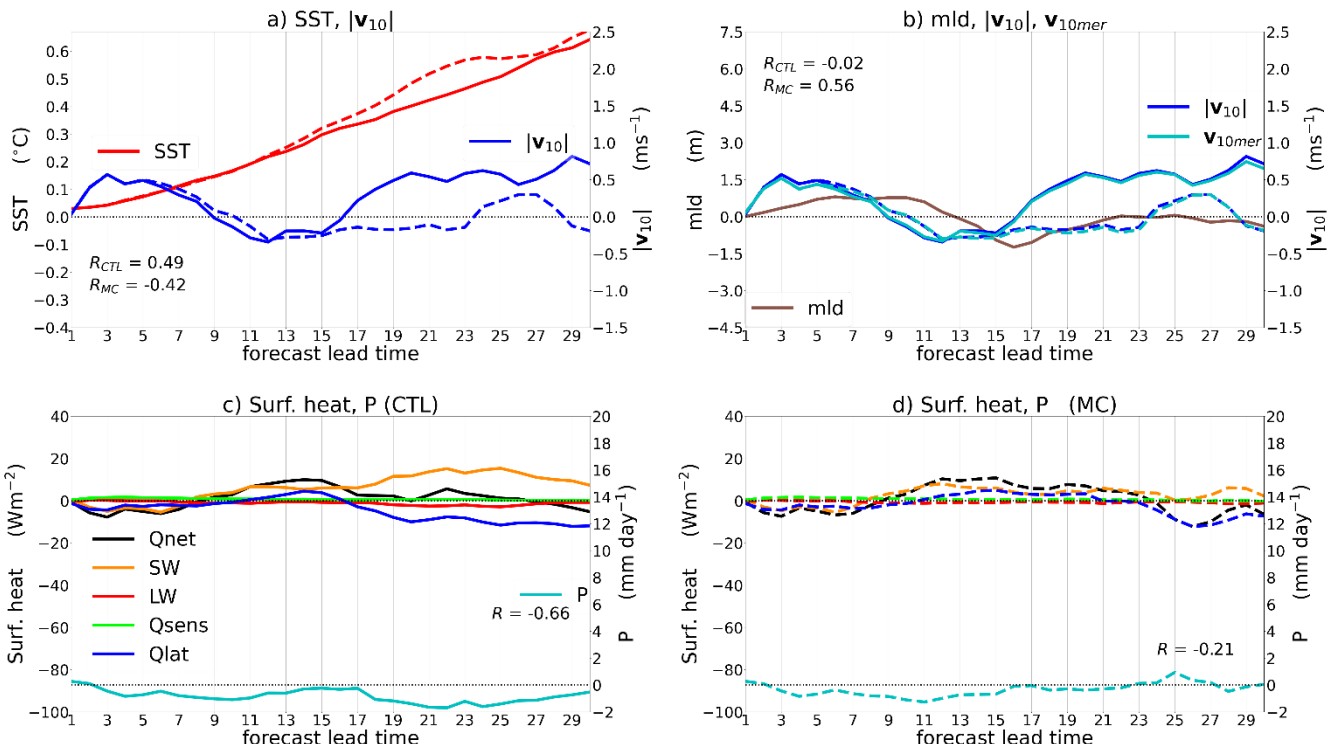

**Figure 14:  Same as Figure 12 but showing biases over the WEIO region (50° - 60° E, 7.5° S – 2.5° N).  In panel (c), the cyan lines now represent the meridional component of 10-wind, v$_{10mer}$.**

### 3.3 Development of ocean errors in the Equatorial Indian Ocean

In order to investigate the possible contribution from the ocean model to the SST errors developing in the EIO region, we now examine longitudinal cross sections of errors in ocean potential temperature and horizontal and vertical currents in the GloSea5 seasonal hindcasts initialised on 25 April. Due to unavailability of the original ocean reanalyses from which the ocean initial conditions for these hindcasts were taken, we approximate these using the day-1 ocean fields from each of the 4 start dates per month of the operational hindcast (i.e., the Control), averaged over the hindcast period (1993-2015), making the reasonable assumption that, during the first day, the modelled ocean does not drift far from the initial state provided by the reanalyses. Figure 15 reveals that, in the Control, there is a contribution to the SST errors from the sub-surface ocean. After the initial surface warming in the easter EIO, related to excessive downward radiative fluxes as discussed in the previous sections, easterly errors in atmospheric low-level winds drive anomalous easterly near-surface ocean currents that help to pull colder water towards the surface in the eastern EIO in the Control, while warmer water from ~100m depth also moves towards the



surface in the west. The errors close to the ocean surface and those at depth are much reduced, though not removed altogether, by nudging the atmosphere over the MC (Fig. 16) and further reduced when the global atmosphere is nudged (Fig. 17), but the basic pattern of warmer in the west and colder in the east remains. The source of the deeper ocean model errors is unknown

and is a current focus of research in the Met Office model development area. It is possible that it partly arises as a response to the initial conditions themselves. Carrying out a set of hindcasts with alternative ocean initial conditions may shed light on this. Alternatively, these errors may have a source outside the region, or deeper in the ocean. Further, detailed, analysis of the development of errors in the ocean component during the seasonal hindcasts is beyond the scope of this paper and will be carried out in future work.


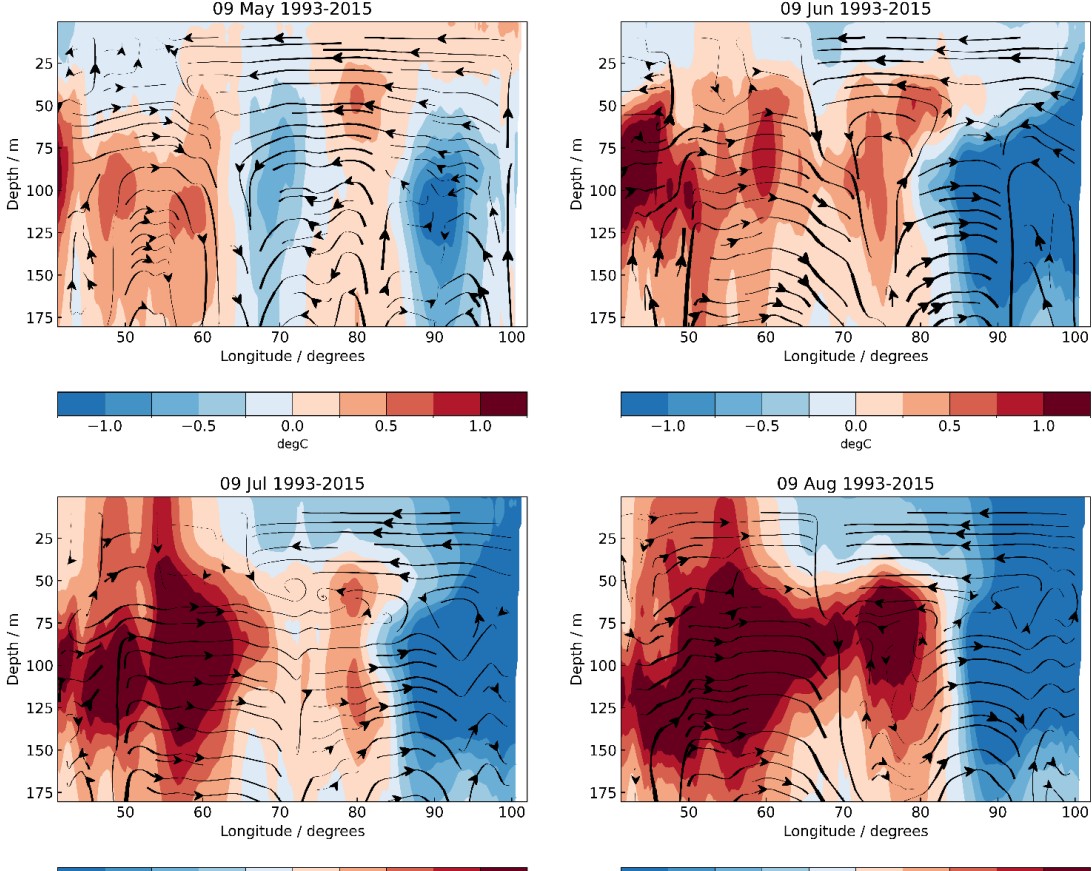

**Figure 15: Differences in sea water potential temperature (shaded), and upward sea water velocity and zonal current (streamlines), between GloSea5 hindcasts and ocean reanalyses at various intervals after initialisation, for Control hindcasts initialised on 25 April 1993-2015.**




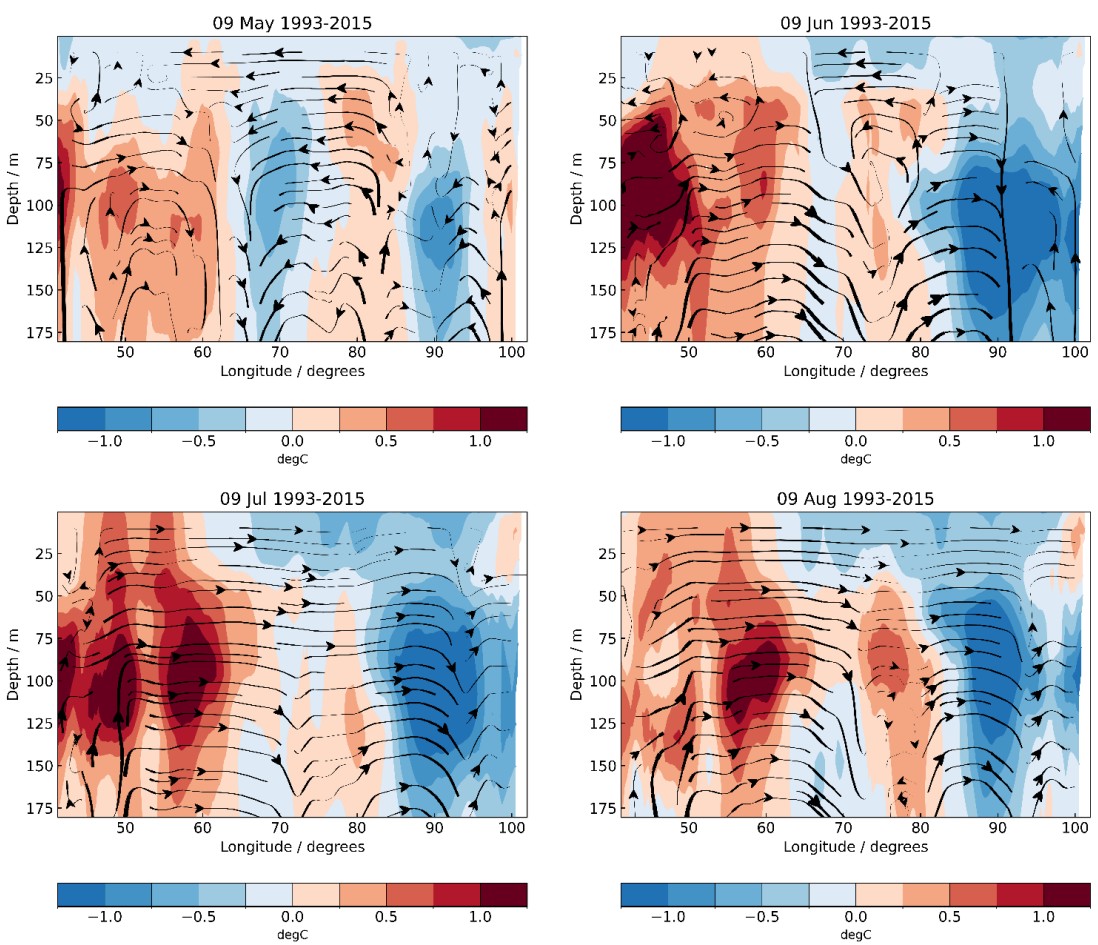

Figure 16: As Fig. 15 but for Nudged MC hindcast.



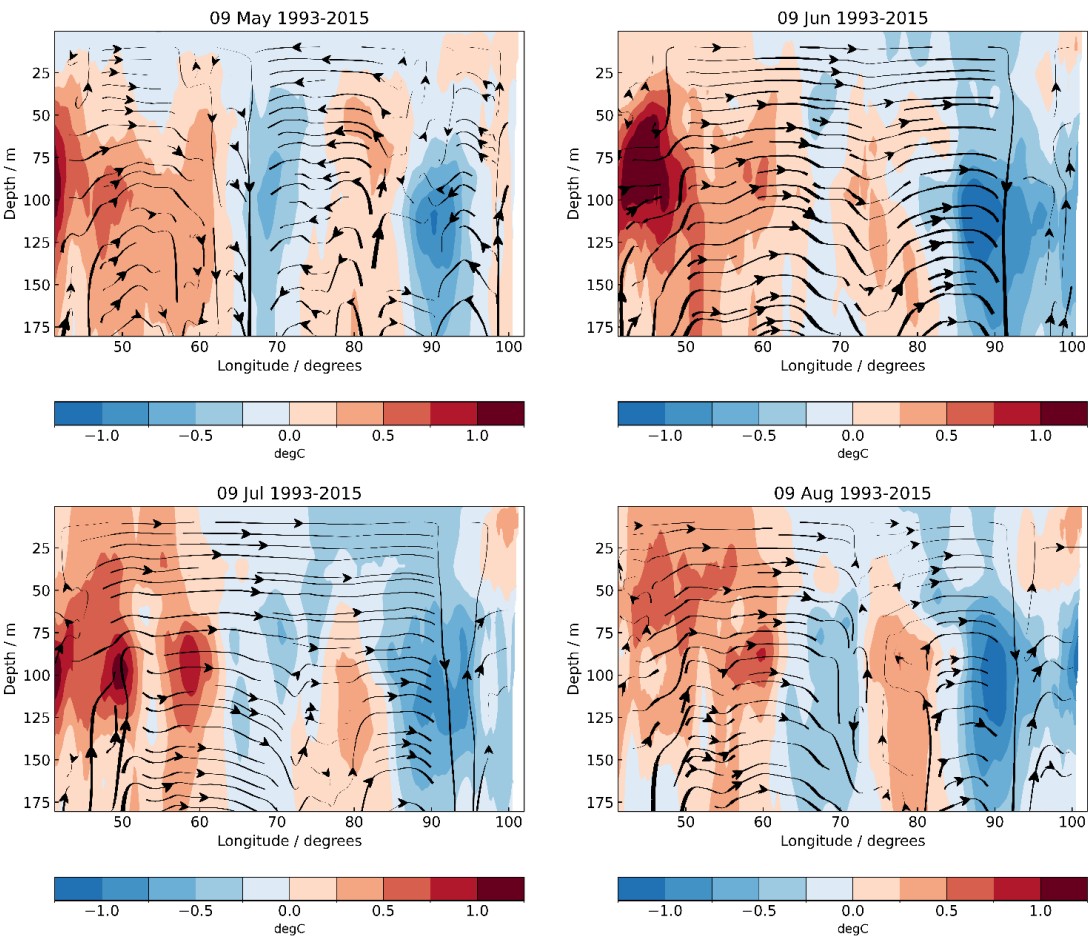

Figure 17: As Fig. 16 but for globally nudged hindcast.

## 4 Summary and conclusions

Martin et al (2021) showed how various techniques and model configurations could be used to shed light on the progression of bias development, their local and remote drivers, and the role of atmosphere-ocean coupling. One of these techniques is the use of regional relaxation ("nudging") experiments to shed light on the contribution from particular regions to more widespread model errors. In order to investigate the contribution from different regions to how ASM errors develop from the very beginning of model simulations, initialised regional relaxation ensemble experiments have been carried out in both coupled NWP and GloSea5 hindcasts. The use of initialised ensembles, carried out for more than two decades in the case of the seasonal hindcasts, allows robust identification of the error patterns arising in the first few days of simulation and their persistence over several pentads beyond.

The results confirm the suggestion of Martin et al. (2021) that errors in the MC region contribute substantially to the EASM circulation errors through their effects on the WNPSH. We also find that errors in the EIO region are associated with circulation



errors over India and the strengthening and extension of the westerly jet across southeast Asia and the SCS into the West Pacific. Locally forced errors over the Indian region also contribute to the EASM errors. Nudging atmospheric winds and temperatures over each of these regions back to reanalyses improves the evolution of the EASMI that is known to characterize the strength of the EASM (Wang et al., 2008).

In both seasonal and coupled NWP model nudged ensemble experiments, we find that the Maritime Continent also plays a role in the development of wind errors that force the SST biases around the Indonesian islands and in the eastern Indian Ocean, indicating an important influence from this highly convective region in the tropical Indian Ocean SSTs in the period after the monsoon transition. The Equatorial Indian Ocean itself plays a key role, however, in the further development of an east-west SST dipole error. Analysis of the development of SST biases in the eastern EIO using CNWP hindcasts shows that the model
first develops a spin-up warm bias, lasting 15 days, that later becomes a long-term cold bias, with a small negative trend. During the spin-up phase, the net surface heat drives the warm bias, whereas the long-term cold bias is forced mainly by excessive surface wind. We hypothesise that atmosphere circulation and temperature errors in the MC and EIO regions combine with an ocean circulation response in a coupled feedback that affects the ASM simulation as a whole; further experiments are planned to investigate this. We also note that the east-west SST dipole error tends to develop even when the
global atmospheric winds and temperatures are nudged back to reanalyses. This suggests that other factors in the atmosphere (such as rainfall and cloud amount affecting surface fluxes) or in the ocean (such as deeper ocean errors) are also contributing to these errors. Analysis of the development of SST biases in the WEIO suggests that that local surface wind speed and surface heat fluxes are not driving the SST bias in this region, and that there is an oceanic transport component to the error. Examination of ocean cross-sections across the EIO confirm that there is a contribution to the SST errors from the sub-surface ocean, and
that errors in atmospheric near-surface winds help to pull colder water towards the surface in the eastern EIO while warmer water from ~100m depth also moves towards the surface in the west. The source of the deeper ocean model errors is unknown, although it is thought that they may partly arise as a response to the ocean initial conditions. In future work, we will investigate nudging other fields in both atmosphere and ocean, and the use of alternative ocean initial conditions, in order to shed more light on the processes driving this important error pattern.

**Code and data availability**

Due to intellectual property right restrictions, we cannot provide the source code for the Met Office Unified Model (MetUM). The MetUM is available for use under licence. For further information on how to apply for a licence, see https://www.metoffice.gov.uk/research/approach/collaboration/unified-model/partnership (last access: 04/01/2024). JULES is available under licence free of charge. For further information on how to gain permission to use JULES for research purposes,
see https://jules.jchmr.org/ (last access: 04/01/2024). The model code for NEMO v3.4 is available from the NEMO Consortium and can be downloaded via http://forge.ipsl.jussieu.fr/nemo/, with software documentation at https://zenodo.org/records/1464817; NEMO System Team (2013), last access: 04/01/2024. The model code for CICE is freely



available from the CICE Consortium, a group of stakeholders and primary developers of the Los Alamos sea ice model, and can be downloaded from the CICE repository (https://github.com/CICE-Consortium/CICE/wiki, last access: 04/01/2024).

Model data used in this study are archived at the Met Office and are available to research collaborators upon request.

**Author contribution**

GMM initiated the study and designed, ran and analysed the nudged seasonal hindcasts. JMR designed, ran and analysed the nudged operational coupled NWP simulations. Both authors contributed to the writing of the manuscript.

**Competing interests**

The authors declare that they have no conflict of interest.

**Acknowledgements**

This work and its contributors were funded by the Met Office Weather and Climate Science for Service Partnership (WCSSP) India project which is supported by the UK Department for Science, Innovation & Technology (DSIT). WCSSP India is a collaborative initiative between the Met Office and the Indian Ministry of Earth Sciences (MoES).

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
