# Peer review of "Using regional relaxation experiments to understand the development of errors in the Asian Summer Monsoon"

_EGUsphere, 2024_

## Author Comment (AC2)

Using regional relaxation experiments to understand the development of errors in the Asian Summer Monsoon
By Gill M. Martin and José M. Rodríguez
MS No.: egusphere-2024-22

Reply to Anonymous Referee #1

This work discusses using a series of regional relaxation or 'nudging' experiments on hindcasts, whereby certain dynamical fields are relaxed back to reanalysis in order to investigate their role on the growth of errors and biases in key components of the Asian Summer Monsoon. The methodology is well motivated and the application of this methodology for studying these errors in coupled NWP and seasonal hindcasts in the Met Office Global Coupled models is new. The results highlight a number of key processes through which errors arise.

This work sheds light on the source of errors and biases in the Met Office Global Coupled models which is important for future model development (that this work will undoubtedly support). This paper fits well within the scope of the journal. The comments I provide for improving the manuscript are about clarity and presentation. Thus I would recommend this article be published after the following comments are addressed.

We thank the Reviewer for their positive comments and their useful recommendations below.

Specific Comments

Introduction

This section provides good motivation for the relaxation methodology but not the wider context of why the specific configurations used in the paper were chosen. It would be of great benefit to the reader to introduce the components of the ASM variability and describing some of the mechanisms through which biases can arise in models as a result of these not being faithfully represented. Specifically it would be useful for the authors to describe the role of each of the regions used in the nudging experiments in controlling the ASM.

As stated in the second paragraph of the Introduction, the regions used were identified in previous work by Martin et al. (2021) from study of the development of errors in the ASM region after initialisation. The error development and the potential role of the regions then chosen for the nudging experiments is reiterated at the start of section 3.1. We have now forward referenced this section in the Introduction to clarify the motivation for the chosen nudging regions at the outset.

There are also many acronyms and areas introduced later on in the paper, some were not defined. For clarity, it would be useful for these to be introduced earlier: the WNPSH, EASMI, the EIO, EEIO and the WEIO.

All of these acronyms are already defined at their first usage: WNPSH (line 119), EASMI (caption of Fig. 1 and line 224), EIO (line 110), EEIO (122) and WEIO (line 220). [Line numbers as in original manuscript.]

Line 44, pg 2, states 'also to increase sample size'. Is sample size referring to deterministic versus ensemble?

Yes, the regional relaxation experiments described in Martin et al. (2021) were deterministic and uninitialized climate runs, so our aim here was to build on this by using ensembles of initialised hindcasts. We have clarified this in the text.

Data description and methods

In the results section there is discussion of a control run, please clarify what this is. Could the authors clarify why the hindcast ensembles are compared to reanalysis, GPCP and OISSTv2 but the Coupled NWP forecasts are compared against FOAM and MetUM operational analysis?

The Control run is the standard operational hindcast, apologies for not making this clear, we have done so in the revision. Yes, the seasonal hindcasts are compared with reanalyses as these are used for initialisation and cover the relevant hindcast period. For Coupled NWP the model is initialised from MetUM op analysis and FOAM ocean initial conditions, hence why this is the comparison. We have now clarified this in section 2.

3.1 ASM errors in seasonal hindcasts

Fig.2 is this the control run?

Yes, we have now confirmed this in the text.

Pg 6, Line 139. It seems a bit strange to comment on the results of the Indonesia and Philippines nudging experiments without showing these. However, if they make a significant contribution to the errors produced by the MC region then it is not necessary for the authors to do so, this seems to be the case but it would be good for the authors to clarify this. This also motivates the authors' assumption that errors in this region are mostly linear.

We did not include plots of the results from Indonesia and Philippines in order to limit the number of figures. We could include these as supplemental figures but we felt that describing the results would be sufficient. The effect of those regions is contributed to the larger MC region so we felt that showing this alone would be sufficient. We have clarified this in the revised text.

3.2 Emergence of Indian Ocean SST errors in coupled NWP hindcasts

Line 284, pg 15 why was 1 ensemble initialised each day, were they different ensemble members, if so, why? Can the authors clarify what the forecast validity time is? Why were

forecasts initialised everyday of 2020 if only certain forecast validity periods between 30 June - 31st August were used?

We apologise for any lack of clarity in this section. Validity time refers to the time at which a forecast is valid, which can be at any time after it is initialised. With a set of forecasts initialised once on every day of the year and each run for 30 days, each of the 63 days in our chosen valid period of 30 June to 31 August will be a valid forecast day within 30 forecasts, but with different lead times according to when they were initialised. In order to collect a representative sample of the errors arising after 1, 2, 3, ... (etc) days of forecast time within our chosen summer monsoon period, we construct 63-member composites of each lead time (time since initialisation) by considering each coupled NWP forecast as a different ensemble member. This approach was also used in Martin et al. (2021).

As explained in Martin et al. (2021), regional biases in the EEIO develop differently for seasonal hindcasts initialised before and after the broadscale seasonal transition, which takes place around mid-May. In this study we focus on biases after the monsoon transition. Although forecasts initialised for every day of 2020 were available, we have only used those forecasts which have valid times within our chosen period, thereby guaranteeing that every forecast used in the composites has been initialised after the transition.

We have changed the text to clarify these points.

Line 289, pg 15. Does EEIO stand for Eastern Eq. Indian Ocean and WEIO western eq. Indian Ocean and do they correspond to box 1 and 2 in Fig 13. If so please specify.

As defined on lines 122 and 220 of the original manuscript, EEIO stands for Eastern Equatorial Indian Ocean and WEIO for Western Equatorial Indian Ocean. As stated on lines 289-290, we then analyse various quantities averaged over the two boxes shown in Fig. 13 that are representative of the different behaviour in the EEIO and WEIO.

There seems to be a rapid change/increase in errors in precipitation, wind, heat flux and SST after day 9. Did atmospheric conditions change on this day perhaps? I.e. If there was low precipitation then there is low variability in the precipitation and hence there would be smaller errors associated with it.

Since what is plotted is an average, for each lead time, of all forecast days between 1 June and 31 August, then the atmospheric conditions would be different on each of these days for any single lead time. Instead, we suggest in lines 305-321 of the original manuscript that the change in error with timescale is due to the different drivers of errors that act on different timescales. We have attempted to clarify this point in the revision.

Figure 12b, 14b there is no dashed red line representing mld in the nudged MC experiment.

We apologise for this omission, which we have found was due to an error in the diagnostic. This has now been corrected and the MLD lines for both experiments are now shown on updated Figs 12 and 14.

**3.3 Development of ocean errors in the Equatorial Indian Ocean**

Line 380, pg 21 is the 'day-1 start day' the 9th of each month? Is this why that date is used in Fig 15, 16 and 17.

Apologies that this was unclear. The operational hindcast has 4 start dates per month, on days 1, 9, 17 and 25 of each month, hence the ocean "initial" state could only be approximated using the daily mean ocean fields from each of these dates. In order to reduce the number of panels on Figs 15, 16 and 17, we chose to illustrate the evolution through the season by plotting the 9th of each month only, but we could choose any of the other dates for those figures. We have clarified this in this section.

Line 385, pg 21 'easter EIO', I assume this is meant to say eastern EIO but is this defined already as EEIO (i.e. box 1 in Fig 13)?

As per our reply to your previous comment, yes, the eastern EIO is the EEIO but it is not defined by the box 1 in Fig 13; that box is merely taken as representative of the EEIO in general. The cross sections in Figs 15 to 17 are averaged between 10°S and 5°N. We note that we failed to state the latitudinal average in the original manuscript, this is now corrected in the text of section 3.3. and the Fig 15 caption.